# Epiphytic Yeasts and Bacteria as Candidate Biocontrol Agents of Green and Blue Molds of Citrus Fruits

**DOI:** 10.3390/jof8080818

**Published:** 2022-08-03

**Authors:** Rania Hammami, Maroua Oueslati, Marwa Smiri, Souhaila Nefzi, Mustapha Ruissi, Francesca Comitini, Gianfranco Romanazzi, Santa Olga Cacciola, Najla Sadfi Zouaoui

**Affiliations:** 1Laboratoire de Mycologie, Pathologies et Biomarqueurs (LR16ES05), Département de Biologie, Université de Tunis-El Manar, Tunis 2092, Tunisia; hammamirania31@gmail.com (R.H.); weslati_marwa@live.fr (M.O.); smiri1990@gmail.com (M.S.); souhaila.nefzi97@gmail.com (S.N.); 2Laboratoire de Biotechnologie Appliquée à l’Agriculture, INRA Tunisie, Université de Carthage, Ariana 2049, Tunisia; mustapha_rssi@yahoo.fr; 3Department of Life and Environmental Sciences, Marche Polytechnic University, Via Brecce Bianche, 60131 Ancona, Italy; f.comitini@staff.univpm.it; 4Plant Pathology, Department of Agricultural, Food and Environmental Sciences, Marche Polytechnic University, Via Brecce Bianche, 60131 Ancona, Italy; g.romanazzi@staff.univpm.it; 5Plant Pathology, Department of Agriculture, Food and Environment, University of Catania, V.S. Sofia, 100-95123 Catania, Italy; olgacacciola@unict.it

**Keywords:** postharvest, orange, lemon, *Penicillium digitatum*, *Penicillium italicum*, *Bacillus amyloliquefaciens*, *Bacillus pumilus*, *Bacillus subtilis*, *Candida oleophila*, *Debaryomyces hansenii*

## Abstract

Overall, 180 yeasts and bacteria isolated from the peel of citrus fruits were screened for their in vitro antagonistic activity against *Penicillium digitatum* and *P. italicum*, causative agents of green and blue mold of citrus fruits, respectively. Two yeast and three bacterial isolates were selected for their inhibitory activity on mycelium growth. Based on the phylogenetic analysis of 16S rDNA and ITS rDNA sequences, the yeast isolates were identified as *Candida oleophila* and *Debaryomyces hansenii* while the bacterial isolates were identified as *Bacillus amyloliquefaciens*, *B. pumilus* and *B. subtilis*. All five selected isolates significantly reduced the incidence of decay incited by *P. digitatum* and *P. italicum* on ‘Valencia’ orange and ‘Eureka’ lemon fruits. Moreover, they were effective in preventing natural infections of green and blue mold of fruits stored at 4 °C. Treatments with antagonistic yeasts and bacteria did not negatively affect the quality and shelf life of fruits. The antagonistic efficacy of the five isolates depended on multiple modes of action, including the ability to form biofilms and produce antifungal lipopeptides, lytic enzymes and volatile compounds. The selected isolates are promising as biocontrol agents of postharvest green and blue molds of citrus fruits.

## 1. Introduction

Citrus fruits are one of the most widely cultivated fruit crops in the world [1]. Citrus fruits are produced in more than 140 countries with a global production of around 158 million tons in 2020 [1]. The interest in this crop is so relevant and continues to increase due to the organoleptic characteristics, health benefits, nutritional value and relatively low price of citrus fruits [2,3,4]. The citrus industry is an economically relevant sector of agriculture in Tunisia, where a land area of around 29 thousand hectares is devoted to citrus growing. In Tunisia, citrus fruits are produced mainly for fresh fruit consumption, and in 2019 the production was estimated at about 393,300 tonnes [1].

A characteristic of citrus fruits is their perishability. During storage, processing, transportation and marketing, considerable losses are caused by fungal rots. In particular, green mold and blue mold incited by *Penicillium digitatum* and *P. italicum*, respectively, are the most damaging postharvest diseases of citrus fruits [5,6,7,8]. *Penicillium* species produce a huge amount of conidia that contaminate citrus fruits in the field as well as in all subsequent postharvest steps of the supply chain. Infections of fruits take place through wounds where the presence of nutrients stimulate conidium germination [5,9]. The infection appears as a soft area surrounding the wound; soon, a white mycelium appears on the lesion and starts producing conidia. The infection leads in a few days to the decay of the entire fruit, which, at a final stage, appears completely covered by conidia, which are green or blue depending on the *Penicillium* species that infected the fruit. Presently, the management of Penicillium rot of citrus fruits has been based on prophylactic practices, cold storage and drenching with synthetic fungicides such as imazalil, thiabendazole, ortho-phenilphenate, trifloxystrobin, azoxystrobin, fludioxonil, cyprodinil and pyrimethanil [10,11,12,13,14,15]. The state-of-the-art on alternative management strategies of green and blue mold of citrus fruits has been recently reviewed [16]. Apart from the toxicological and environmental implications, the intensive use of synthetic fungicides has led to the proliferation of fungicide-resistant populations with the consequent reduction in effectiveness in disease control [17]. These drawbacks of synthetic fungicides have prompted researchers to seek alternative and safer methods for controlling the postharvest decay of fruits [6,18,19,20,21,22,23]. The application of antagonistic bacteria and yeasts, natural inhabitants of the surfaces of vegetables and fruits, is a promising alternative to synthetic fungicides [24,25]. In general, the multiple modes of action of biological control agents (BCAs) makes pathogen resistance unlikely [26]. Specifically, yeasts possess characteristics that make them interesting as BCAs, including simple nutritional requirements, rapid growth and proliferation, tolerance towards some pesticides, capability to colonize and survive on fruit surfaces for long periods as well as to compete for space and nutrients [27,28,29]. The biocontrol activity of antagonistic yeasts can be related to several mechanisms, including the ability to adhere on specific sites of both host and pathogen cells, competition with the fungal pathogen for nutrients or space, direct parasitism and quorum sensing, secretion of lytic enzymes and antimicrobial substances, production of volatile organic compounds (VOCs), biofilm formation on the inner surface of wounds, extracellular proteic toxins production and induction of host resistance [29]. Among these mechanisms, competition for nutrients and space seems to be the prevalent action mechanism of many yeasts. Several antagonist yeasts have been reported to effectively control the postharvest decay of citrus fruits [30], including *Candida oleophila* [31,32], *C. saitoana* [33], *Debaryomyces hansenii* [34,35], *Kloeckera apiculata* [36], *Metschnikowia pulcherrima* [37], *Cryptococcus laurentii* [38,39], *Wickerhamomyces anomalus* (formerly, *Pichia anomala*) [28,40], *Pichia guilliermondii* [41,42], *P. galeiformis* [43], *Meyerozym aribbica* (anamorph *Candida fermentati*) [44], *Aureobasidium pullulans* [45], *Rhodotorula glutinis* [46] and *Hanseniaspora uvarum* Y3 [47]. In addition to yeasts, non-pathogenic bacteria of the genus *Bacillus* were reported to be effective as BCAs against postharvest decays of citrus fruits, including *Bacillus subtilis* [48,49,50,51,52], *B. pumilus* [52], *Paenibacillus polymyxa* [53] and *B. amyloliquefaciens* [54,55,56]. In general, *Bacillus* species have shown a high capability to colonize the plant surface and sporulate [57]. Moreover, they exert an antagonistic activity against a wide range of plant pathogens [58]. The efficacy of most *Bacillus* species as BCAs depends on their ability to produce enzymes and other compounds with high antibiotic activity, such as cyclic lipopeptides (CLPs), e.g., iturin, surfactin and fengycin [59], which modify the permeability of the cell membrane of postharvest fungal pathogens [60]. In addition to CLPs compounds, *Bacillus* species synthetize several enzymes such as chitinases, β-1,3-glucanases and proteases [61,62] that degrade fungal polymers, as well as volatile organic compounds (VOCs) that induce plant resistance to biotic stresses.

The aims of the present study were to (i) screen epiphytic yeasts and bacteria of the genus *Bacillus* for their ability to antagonize *P. digitatum* and *P. italicum*, (ii) identify the most effective candidate BCAs among the microorganisms tested, (iii) investigate their potential application to prevent Penicillium rot of orange fruits, (iv) evaluate the effects of treatments with candidate BCAs on the quality of orange fruits during storage and (v) elucidate the mechanism(s) of action of these candidate BCAs.

## 2. Materials and Methods

### 2.1. Citrus Fruits

Mature orange (*Citrus* × *sinensis* ‘Valencia late’) and lemon (*Citrus* × *lemon* ‘Eureka’) fruits without any visible symptom, selected based on homogeneity of color and size, were used throughout this study.

### 2.2. Fungal Pathogens

*Penicillium digitatum* (K7) and *P. italicum* (K2) isolates used in this study were from the fungal culture collection of the Mycology, Pathologies and Biomarkers Laboratory (LMPB), Faculty of Sciences, University of Tunis El Manar, Tunis, Tunisia. They were routinely cultured on Potato Dextrose Agar medium (PDA, Biolife, Milan, Italy) with periodic transfers on healthy citrus fruits surface disinfested with 2% sodium hypochlorite (NaClO) and subsequent re-isolation to maintain their pathogenicity.

### 2.3. Epiphytic Yeasts and Bacteria

*Bacillus* and yeast isolates were recovered from healthy fruits of several citrus varieties. The isolation of yeasts was performed with the serial dilution method described by Chalutz and Wilson (1990) [63]. The same serial dilution was used to isolate *Bacillus* strains after a heat treatment for 30 min at 80 °C in a water bath (Memmert GmbH + Co.KG, Schwabach, Germany) [64]. Aliquots of 100 µL were poured in Petri dishes onto Tryptic Soy Agar medium (TSA, Biolife, Milan, Italy). Dishes were incubated for 3 days at 28 °C in the dark. Representative colonies were randomly selected and streaked onto fresh TSA dishes to obtain single colonies. Isolates were maintained at −80 °C for long-term storage.

### 2.4. In Vitro Screening of Isolates

In vitro antagonism assays against *P. digitatum* and *P. italicum* were performed using the dual-culture test described by [65]. Each yeast or bacterium was applied as a straight line passing through the center of 9 cm Petri dish containing PDA medium. Then, a mycelium plug of 5 mm in diameter of the pathogen (*P. digitatum* or *P. italicum*) was placed approximately 2.5 cm away from each side of the tested strain. The dishes were incubated at 25 °C for 7 days. Dishes inoculated only with green (*P. digitatum*) or blue (*P. italicum*) molds were used as a control. The antagonistic activity was expressed as percent inhibition of mycelium growth compared to the control. Antifungal activity was calculated according to the following formula: I (%) = (R1 − R2)/R1 × 100 (where I is the percentage of growth inhibition, R1 stands for the mean radius of *Penicillium* spp. colony with no bacterium or yeast and R2 indicates the colony area of *Penicillium* spp. grown in the presence of bacterium or yeast to be tested). Each selected isolate was tested three times, with three replicates for each experiment [66].

### 2.5. Antagonistic Activity on Citrus Fruits

Bacterial and yeast isolates selected in vitro on the basis of their inhibitory activity on mycelial growth of *P. digitatum* and *P. italicum* were also tested in vivo. The antagonistic activity of isolates was tested on citrus fruits wound-inoculated separately with *P. digitatum* and *P. italicum* according to protocols described by [31]. Detached fruits of ‘Valencia late’ sweet orange and ‘Eureka’ lemon were disinfected for 2 min in 2% NaClO and then washed with sterile distilled water (SDW). Then they were wounded in equatorial position (a single wound/fruit, approximately 3 mm in depth × 3 mm in width) using a sterile needle. Each wound was inoculated with 30 µL of cell suspension of antagonist microorganism adjusted to a concentration of 10^8^ CFU/mL. After 24 h, each wound was inoculated with 20 µL of an aqueous suspension of *P. digitatum* or *P. italicum* (10^5^ conidia/mL). Control fruits were wounded, and after 24 h 30 µL of SDW was pipetted on the wound. After air-drying, the inoculated fruits were kept in plastic boxes. Boxes were kept in a growth chamber at two different incubation conditions, ambient temperature (around 24 °C) with high humidity (95% R.H.) for 7 days and 4 °C for 30 days. Four plastic boxes (15 cm × 25 cm), with three fruits per box, were used per treatment. Each treatment comprised four replicates (12 fruits/replicate), and the experiment was repeated twice. In a first set of experiments, the efficacy of the five selected candidate BCAs in preventing the infection of *P. digitatum* and *P. italicum* on orange fruits kept at room temperature was compared. In a second series of experiment, the efficacy of the five selected candidate BCAs in preventing the infections of *P. digitatum* on orange and lemon fruits, respectively, was compared. In a third series of experiments, the efficacy of the candidate BCAs in preventing the infections of *P. digitatum* and *P. italicum* on orange fruit kept at 4 °C for a long time (30 days) was evaluated. In a fourth series of experiments, the effect of storage conditions (7 days at room temperature and 30 days at 4 °C) on the effectiveness of candidate BCAs in controlling *P. digitatum* infections on lemon fruit was evaluated. The lesion diameter (LD) and the disease reduction (DR) were recorded. Lesion diameter (LD) was computed as mean of longest and shortest diameters of the fruit lesion; fruits with LD greater than 3 mm (the size of wound) were considered decayed, and the disease reduction (DR) was calculated by applying the following formula described by [67]:DR (%) = (a − b)/b × 100
where DR is the percentage of disease reduction, a is the lesion diameter in control fruit (cm) and b is the lesion diameter in treated fruits (cm).

### 2.6. Molecular Identification of Yeast and Bacterial Isolates and Phylogenetic Tree Construction

Two yeast isolates (L12 and L16) were identified by sequencing the internal transcribed spacer (ITS) regions of the nuclear ribosomal DNA (rDNA). DNA was extracted using the Wizard Genomic DNA kit (Promega Corporation, Madison, WI, USA) following the instructions of the manufacturer. The DNA ITS regions were amplified by PCR with the primers ITS1-F (5′-TCCGTAGGTGAACCTGCGG-3′) and ITS4-R (5′-TCCTCCGCTTATTGATATGCGC-3′), as previously described by [68].

Phylogenetic identification of bacteria was based on the 16S rRNA gene sequences. The 16S rRNA gene-encoding region was PCR-amplified using the universal primer pair forward S1 (5′-AGAGTTTGATCMTGGCTCAG-3′) and reverse S2 (5′-GGMTACCTTGTTACGAYTTC-3′) [69]. Nucleotide sequences were blasted and aligned using the NCBI database. Phylogenetic tree of strain was constructed using the neighbor-joining method with the software MEGA X (Molecular Evolutionary Genetic Analysis version 11). The MEGA project was taken over by Kumar at Temple University and Tamura at Tokyo Metropolitan University [70].

### 2.7. Effectiveness of Isolates in Reducing Orange Fruit Decay

The effectiveness of selected bacterial (S15, S57 and S67) and yeast (L12 and L16) isolates in preventing natural infections of *Penicillium* species on orange fruits was assessed according to Lai et al. (2012) [53], with little modifications. An experimental trial was carried out in controlled atmosphere under storage conditions similar to those of commercial packinghouses. ‘Valencia Late’ orange fruits were dipped into a cell suspension (10^8^ cells/mL) of the selected yeasts and bacteria for 5 min. Non-treated fruits were used as control. In addition, fruits treated with imazalil (2 mL/L a.i.) were included as a reference to simulate routine chemical treatment to prevent *Penicillium* rot in commercial packinghouses. A sulfate solution of imazalil (7.5% a.i.) was used for fruit treatment. Overall, 186 fruits (31 × 6 replicates) were used for each treatment. Fruits were dipped in the bacterial/yeast suspension or imazalil for 2 min, air-dried and stored in plastic boxes. Boxes were incubated for 90 days in a cold chamber at 4 °C and 90–95% RH in the dark. At the end of the storage period, fruits were visually checked, and the proportion of rotten fruits for each treatment was recorded. The DI was determined as the number of rotten oranges divided by the number of total oranges × 100, while DS was determined according with the formula proposed by [71]:DS (%) = ∑ (n × r1) … (n × r5)/5N) × 100 
where n = number of decayed fruits in each class; r1 … r5 = numerical values of classes; and N = total number of fruits multiplied by the maximum numerical disease class.

The experiments were repeated over three consecutive years (2018, 2019 and 2020), and data were combined.

### 2.8. Effect of Treatments on Postharvest Quality of Oranges

The effect of treatments with antagonistic bacterial and yeast isolates on postharvest quality of orange fruits was evaluated by determining the following parameters: weight loss (%), fruit firmness (N), total soluble solids (%), titratable acidity (%) and ascorbic acid content (mg/100 g). Fruits were treated with the candidate BCAs by dipping and stored at 4 °C for 90 days, as described previously. Each treatment comprised three replicates with 10 fruits per replicate.

Weight loss was assessed on 10 randomly selected fruits from each replicate at the end of the storage period (90 days). The weight loss was calculated using the following formula: weight loss (%) = (initial weight of fresh fruit − weight of fruit after storage)/initial weight of fresh fruit × 100 [72]. Fruit firmness (N) of each orange fruit was measured at two opposite points of the equatorial region by using a firmness tester (Gy-4, Zhejiang Top Instrument Co. Ltd., Hangzhou, China) fitted with a 5 mm diameter probe [73]. Total soluble solids (TSSs) content in the juice was determined by measuring the refractive index with a hand refractometer (PR-32, Atago Co., Tokyo, Japan) at room temperature. Results were expressed as percentage Brix [74]. Titratable acidity (TA) was measured by titration of 10 mL of fruit juice with 0.1 M NaOH to pH 8.3 and phenolphthalein as an indicator. Results were expressed as percentage of citric acid [75]. The ascorbic acid (Vit C) was estimated using the 2,6-dichlorophenol-indophenol titrimetric method according with [72].

### 2.9. Mechanisms of Action of Isolates Selected as Candidate BCAs

#### 2.9.1. Hydrolytic Enzymatic Activity of Isolates Selected as Candidate BCAs

Production of extracellular hydrolytic enzymes, including protease, amylase, cellulase, chitinase, pectinase, mannanase and urease, were tested by spot-inoculating the candidate BCAs in Petri dishes on the appropriate medium: skim milk [28], starch [76], carboxymethyl cellulose [77], chitin [28], pectin [78], locust bean gum [79] and urea-indole medium [80], respectively. Three replicate dishes were used for each isolate. Development of a halo surrounding the colony was considered as positive result.

#### 2.9.2. Ability of Isolates Selected as Candidate BCAs to Produce Volatile Organic Compounds

A double Petri dish assay was used to test the antifungal activity of volatile organic compounds (VOCs) produced by yeast and *Bacillus* isolates against *P. digitatum* and *P. italicum* according to [28]. PDA dishes with only *Penicillium* species served as control. The radial growth inhibition of test fungi was measured as previously described. For each treatment, three replicate dishes were used, and the experiment was performed twice.

#### 2.9.3. Biofilm Formation

Biofilm formation by yeast and bacterial isolates selected as candidate BCAs was evaluated with the microtiter dish assay using the dye crystal violet as described by [28,81].

#### 2.9.4. PCR Detection of Genes Encoding for Antibiotic Biosynthesis

Bacterial isolates were screened for their ability to produce lipopeptides, using different pairs of specific primers that amplify genes encoding for fengycin (FENDF, 3′-GGCCCGTTCTCTAAATCCAT-5′; FENDR, 5′-GTCATGCT-GACGAGAGCAAA-3′), bacillomycin (BMYBF, 3′-GAATCC-CGTTGTTCTCCAAA-5′; BMYBR, ‘5-GCGGGTATTGAAT-GCTTGTT-3′), bacilysin (BACF, ‘3-CAGCTCATGGGAAT-GCTTTT-5′; BACR, ‘5-CTCGGTCCTGAAGGGACAAG-3′), surfactin (SRFAF, ‘3-TCGGGACAGGAAGACATCAT-5′; SRFAR, ‘5-CCACTCAAACGGATAATCCTGA-3′), iturin A (ITUD1F, ‘3-GATGCGATCTCCTTGGATGT-5′; ITUD1R, ‘5-ATCGTCATGTGCTGCTTGAG-3′) and mycosubtin (Am1-F,’3-CAKCARGTSAAAATYCGMGG-5′; Tm1-R, ‘5 CCDASATCAAARAADTTATC-3′). PCR reactions were performed in 25 µL of reaction mixtures containing 1.5 µL of DNA (20–30 ng/µL), 1.5 µL of each primer (20 µM) and 12.5 µL of PCR Master Mix. The amplification were performed using a DNA thermal cycler (Biometra GmbH, Göttingen, Germany) with the following cycle conditions: an initial denaturation step at 95 °C for 4 min, followed by 40 cycles each at 94 °C for 1 min, 58 °C for 1 min and 70 °C for 1 min; and a final extension at 70 °C for 5 min for fengycin (FEND-F/FEND-R), bacilysin (BAC-F/BAC-R) and surfactin (SRFA-F/SRFA-R). An initial denaturation at 95 °C for 4 min followed by 40 cycles each at 94 °C for 1 min, 55 °C for 1 min and 70 °C for 1 min; and a final extension at 70 °C for 5 min for bacillomycin (BMYB-F/BMYB-R). An initial denaturation at 94 °C for 3 min followed by 35 cycles each at 94 °C for 1 min, 60 °C for 30 s, and 72 °C for 1 min 45 s; and a final extension at 72 °C for 6 min for iturin A (ITUD1-F/ITUD1-R). An initial denaturing step at 94 °C for 3 min, followed by 30 cycles at 94 °C for 1 min, a hybridization of 30 s at 43 °C for Am1-F/Tm1-R an elongation at 72 °C for 2 min and, finally, a final extension step of 10 min at 72 °C for mycosubtilin (Am1-F/Tm1-R) [82,83,84,85].

### 2.10. Statistical Analysis

Statistical analysis was performed using the program Stat soft. Inc. 2011. STATISTICA (data analysis software system) Version 10, www.statsoft.com, accessed on 21 April 2021. An analysis of variance and mean comparison test were performed using Duncan’s test with 5% significance level. A completely randomized design was used for all experiments, with three replications for each treatment. The data presented are from representative experiments that were repeated twice with similar results.

## 3. Results

### 3.1. Screening of Microorganisms for In Vitro Antagonistic Activity

Altogether, 180 isolates (150 bacteria and 30 yeasts) obtained from the fruits of ‘Eureka’ lemon, ’Washington Navel’ and ‘Valencia-Late’ sweet orange; ‘Star Ruby’ grapefruit; and ‘Clementine’ tangerine from different orchards in the Cap Bon area (Tunisia) were screened for their ability to inhibit *Penicillium digitatum* isolate K7 and *P. italicum* isolate K2. Only 70 isolates (60 bacteria and 10 yeasts) showed a significant antifungal activity, with variable efficacy (data not shown). Five isolates, three bacteria (S15, S57 and S67) and two yeasts (L12 and L16) were the most effective in inhibiting the growth of both pathogens (Table 1).

In vitro tests showed that the three bacterial isolates produced clearly distinct inhibition areas (Figure 1). The highest antifungal activity was shown by the isolate S15, which inhibited the *P. digitatum* growth by 76%, while the highest inhibitory activity on *P. italicum* (73% reduction in mycelium growth) was shown by the isolate S57. Compared with bacteria, yeasts showed a lower inhibitory activity on the mycelial growth of both *P. digitatum* and *P. italicum*. In particular, the isolate L12 inhibited the mycelium growth of *P. digitatum* and *P. italicum* by 32 and 20%, respectively, while the isolate L16 inhibited the mycelium growth of *P. digitatum* and *P. italicum* by 28 and 20%, respectively.

### 3.2. Bioassay on Citrus Fruits

The efficacy of the five selected yeast and bacterial isolates as candidate BCAs was evaluated by the assay on wound-inoculated citrus fruits. On orange fruits, significant differences were observed among the candidate BCAs tested in their effectiveness in preventing the infections by the two *Penicillium* species. The yeast isolates L12 and L16 and the *Bacillus* isolate S15 were the most effective against *P. digitatum*, while the yeast isolate L12 and the *Bacillus* isolate S57 were the most effective against *P. italicum* (Figure 2). No significant difference was observed between the efficacy of yeast isolate L12 against *P. digitatum* and *P. italicum*. This isolate was the most effective among the tested candidate BCAs and reduced the disease caused by the two *Penicillium* species by 95 and 92%, respectively (Figure 2). Similarly, no significant difference was observed between the effectiveness of the Bacillus isolate S57 in inhibiting the orange fruit infections by *P. digitatum* or *P. italicum*, respectively. By contrast, the other two *Bacillus* isolates and the yeast isolate L16 were significantly more effective in inhibiting the infections of *P. digitatum* than those of *P. italicum* (Figure 2). Isolate S15 was as effective as the isolate L12 in inhibiting the infections of *P. digitatum* on orange fruits stored at room temperature but was significantly less effective in inhibiting the infections of *P. italicum* (Figure 2).

The isolate L12 was also the most effective in reducing the infections of *P. digitatum* on lemon fruits at room temperature (Figure 3). The isolates L12, L16, S57 and S67 showed effectiveness on lemon fruits comparable to that observed on orange fruits. The *Bacillus* isolate S15 was an exception as, at room temperature, it was significantly less effective on lemon fruits: 41.58% vs. 66.02% disease reduction, respectively (Figure 3). Among the bacterial isolates, the isolate S57 was the most effective on lemon fruits at room temperature (Figure 3).

Interestingly, after long storage at +4 °C the *Bacillus* isolate S15 was as effective as the yeast isolate L12 in controlling green mold on orange fruits (Figure 4).

Similarly, at 4 °C, the *Bacillus* isolates S15 and S57 were as effective as the yeast isolates in controlling green mold on lemon fruits (Figure 5).

### 3.3. Identification of Selected Microorganisms

Yeast and bacterial isolates were identified by ITS rDNA and 16S rDNA sequences, respectively (Figure 6 and Figure 7). The results showed that yeast isolate L12 was closely related to *Candida oleophila* JX188107 with 98.60% similarity, and isolate L16 was closely related to *Debaryomyces hansenii* KJ507663 with 99.30% similarity.

Based on a partial 16S rDNA sequence analysis, all three bacterial isolates were assigned to the genus *Bacillus*, and more in detail, the isolate S15 showed a high similarity (˃99%) with *B. amyloliquefaciens* (MK182997) and the isolate S57 and S67 showed a high similarity with *B. subtilis* (97%) and the *B. pumilus* (99.63%) reference isolates, KU904283 and MK430437, respectively. Nucleotide sequence data of the two yeasts and three selected bacteria were deposited in the GenBank database under the accession numbers MZ723967, MZ724412, MZ724619, MZ724620 and ON528702, respectively.

### 3.4. Decay Due to Natural Infections

The results of experiments aimed at investigating the efficacy of selected BCAs in preventing natural infections of postharvest molds in ‘Valencia late’ oranges are shown in Figure 8. All treatments significantly reduced (*p* < 0.05) the disease incidence and severity compared with the non-treated control. After storage at +4 °C for 90 days, the untreated control showed a decay incidence 59.28%, whereas the decay incidence in orange fruits treated with candidate BCAs was significantly lower, ranging from 21.86% to 45.15%. The most effective in reducing the incidence of fruit decay was the treatment with *B. amyloliquefaciens* S15, which proved to be even more effective than the conventional treatment with imazalil. The disease incidence in fruits treated with *B. pumilus* S67, *C. oleophila* L12 and *B. subtilis* S57 did not differ significantly from the incidence of disease in fruits treated with *B. amyloliquefaciens* S15 or in those treated with imazalil. The least effective BCAs in reducing the incidence of postharvest decay of orange fruits was the isolate L16 of *D. hansenii*, whose efficacy, however, was comparable to that of imazalil (Figure 8a).

Although all tested yeasts and bacteria significantly affected some quality parameters of treated ‘Valencia late’ orange fruits (Table 2), none of them impaired the marketability and shelf life of fruits. Conversely, after storage at +4 °C for 90 days, fruit treated with candidate BCAs showed a significantly lower weight loss and higher ascorbic acid content than non-treated control fruit.

### 3.5. Hydrolytic Enzymes

In vitro assays for the ability to produce hydrolytic enzymes revealed a great variability among the isolates tested (Table 3). All isolates were able to hydrolyze mannanase and unable to produce urease. *Bacillus amyloliquefaciens* isolate S15 showed the broadest spectrum of enzymatic activities, including amylase, protease, cellulase, chitinase and pectinase in addition to mannanase. *B. subtilis* isolate S57 produced all these enzymes except chitinase, while *B. pumilus* isolate S67, other than mannanase, produced only protease. Both yeasts, *C. oleophila* isolate L12 and *D. hansenii* isolate L16, produced cellulase. In addition, isolate L16 produced pectinase.

### 3.6. Effect of VOCs Produced by Candidate BCAs

All five selected candidate BCAs produced VOCs. VOCs produced by *Bacillus* spp. isolates S15, S57 and S67 showed a significant inhibitory effect on the mycelial growth of both *P. digitatum* and *P. italicum.* The *B. amyloliquefaciens* S15 isolate inhibited the mycelium growth of these two fungi by 61 ± 0.01 and 54 ± 0.00%, respectively; the *B. subtilis* S57 isolate by 60 ± 0.00 and 51 ± 0.05%, respectively; and the *B. pumilus* S67 isolate by 47 ± 0.02 and 51 ± 0.01%, respectively (Table 4). VOCs produced by the two selected yeast isolates showed a lower inhibitory effect on the mycelium growth of the two test fungi. Moreover, VOCs produced by isolate L12 of *C. oleophila* were more effective against *P. italicum* than against *P. digitatum* (mycelium growth inhibition 41 ± 0.01 and 7 ± 0.05%, respectively). By contrast, VOCs produced by isolate L16 of *D. hansenii* showed a higher inhibitory effect against *P. digitatum* than against *P. italicum* (mycelium growth inhibition 41 ± 0.01 and 6 ± 0.00%, respectively).

### 3.7. Biofilm Formation

The results of tests aimed at evaluating the ability of candidate BCAs to form a biofilm are reported in Table 5. After 3 h of incubation, *C. oleophila* showed the highest ability to form a biofilm (OD 0.91 ± 0.28), followed by *D. hansenii* (OD 0.75 ± 0.25). The bacterial isolates, *B. amyloliquefaciens* S15, *B. subtilis* S57 and *B. pumilus* S67, showed a significantly lower ability to form a biofilm (OD 0.20 ± 0.35, 0.26 ± 0.39 and 0.29 ± 0.00, respectively).

### 3.8. Detection of Genes Encoding for Antibiotics

PCR amplification using six degenerate primer pairs was performed to detect the presence of genes implicated in biosynthesis of fengycin, surfactins, mycosubtilin, bacillomycin and bacilysin. Five genes encoding for the lipopeptides fengycin, surfactin, iturin A, bacillomycin and bacilysin were detected in all the candidate BCAs bacteria tested (*B. amyloliquefaciens* S15, *B. subtilis* S57 and *B. pumilus* S67) (data not shown). However, the presence of the mycosubtlin-encoding gene was only detected in *B. amyloliquefaciens* S15 (Figure 9).

## 4. Discussion

In this study, 5 out of a total of 180 yeast and bacterial isolates, recovered from the peel of citrus fruits in Tunisia, were selected for their ability to antagonize *P. digitatum* and *P. italicum*, the causal agents of green and blue molds of citrus fruits, respectively. The rationale of selecting candidate BCAs from Tunisian populations of microorganisms inhabiting the peel of citrus fruit was the assumption that endemic epiphytic isolates are inherently better adapted to local environmental conditions, which would confer them a higher carposphere competence and consequently a better performance as BCAs of sympatric pathogens [41,86,87]. The five yeast and bacterial isolates were originally selected on the basis of their ability to inhibit the mycelium growth of *Penicillium* species in dual culture tests, and although the in vitro antifungal activity was not directly correlated with the in planta efficacy as BCAs, they were also proven to be highly effective in preventing post-harvest green and blue molds on citrus fruits in different storage conditions. The two selected yeast isolates were identified as *C. oleophila* and *D. hansenii* and the tree bacterial isolates as *B. amyloliquefaciens*, *B. pumilus* and *B. subtilis* by sequencing the ITS rDNA and the 16S rRNA gene regions, respectively. Several studies have shown considerable potential of beneficial microorganisms retrieved from diverse agro-systems or natural environments that can be integrated into plant disease management strategies. In particular, while soil is the main source of beneficial bacteria such as *Pseudomonas* and *Bacillus* species or filamentous fungi of the genus *Trichoderma*, widely used for soil bioremediation or as BCAs against soilborne plant diseases [88,89,90,91,92,93], fruit surfaces are the most appropriate source of epiphytic, carposphere-competent microbial antagonists able to compete with postharvest fruit pathogens [30,94]. The use of these microbial antagonists as BCAs has the advantage of being a toxicologically and environmentally safe approach to control diseases of fruit crops [95,96]. All five isolates selected in this study showed a relevant inhibitory activity against *P. digitatum* and *P. italicum*, both in vitro and in vivo. A significant antagonistic bacterial isolates × *Pencillium* species interaction was observed in in vitro assays and was also confirmed in tests aimed at evaluating the effectiveness of these isolates in preventing Penicillium molds on fruits. In general, the three selected bacterial isolates were more effective on *P. digitatum* than on *P. italicum*. Several species of *Bacillus* have been previously reported to be effective biocontrol agents of postharvest fungal pathogens, such as *Monilinia* spp., *Botrytis cinerea* and *Penicillium* spp. [30,97]. The *B. amyloliquefaciens* isolate selected in this study showed a high in vitro inhibitory activity on the mycelim growth of *P. digitatum* and proved to be very effective in preventing Penicillium mold of lemon fruits stored at low temperature (4 °C). These results are consistent with those of Calvo et al. (2017) [54], who reported a strong in vitro antifungal activity of *B. amyloliquefaciens* BUZ-14 against many postharvest pathogens including *M. fructicola*, *B. cinerea*, *M. laxa*, *P. expansum*, *P. italicum* and *P. digitatum*. Similarly, Deng et al. (2020) [94] reported that *B. sonorensis* KLBC GS-3 exhibited a significant inhibitory in vitro activity against *P. digitatum*. The potential of *B. amyloliquefaciens* as a biocontrol agent of green mold of citrus fruits caused *P. digitatum* was previously demonstrated by several other researchers [74,98]. In addition, the yeast *C. oleophila* that in this study showed a strong inhibitory activity against *P. digitatum* and *P. italicum* and effectively prevented fruit rot incited by these fungi is the base of Aspire^®^, a commercial product registered for the control of postharvest decay of citrus and pome fruit [99]. It was demonstrated that the modes of action of Aspire^®^ included, among others, nutrient competition, site exclusion and direct mycoparasitism. The results of this study confirm that the antifungal efficacy of the five yeast and bacterial isolates selected in this study depends on multiple modes of action. In particular, marked inhibition zones were observed between the colonies of both *P. digitatum* and *P. italicum* and the selected bacterial isolates in in vitro dual cultures, indicating that the antifungal activities exhibited by these candidate BCAs was due to extracellular diffusible metabolites such as hydrolytic enzymes and antibiotics. The inhibition zones induced by the three selected *Bacillus* isolates in the dual culture tests were significantly larger than those induced by yeast isolates. The three bacterial isolates were therefore selected as target candidate BCAs for further tests aiming at evaluating the ability to produce antibiotics. PCR amplification using degenerate primer pairs revealed the presence of the genes encoding for the amphiphilic cyclic lipopeptides (CLPs) fengycin, surfactin, iturin A, bacillomycin and bacilysin in all three isolates and of the gene involved in the mycosubtlin synthesis in the *B. amyloliquefaciens* isolate, suggesting that very probably antibiotics are at least in part responsible for the antifugal activity of these bacteria. It is known that CLPs, especially the members of the iturin and surfactin families are able to modify the permeability and lipidic composition of fungal membrane [100]. Members of the iturin family, including iturin A, bacillomycin and mycosubtilin, have been reported to possess strong antifungal activity against a wide range of fungi, including important plant pathogens [101,102]. In particularly, it was reported that the ability of bacteria in the genus *Bacillus* to produce iturin A conferred them antifungal activity against the postharvest pathogens of citrus [103]. Similarly, the antifungal activity of some *B. amyloliquefaciens* strains against diverse plant pathogens depended on the production of iturin [104,105]. In vitro assays for the ability to produce enzymes revealed substantial differences in the spectrum of enzymatic activities between *Bacillus* and yeast isolates selected in this study. In previous studies, it was demonstrated that *B. subtilis*, which is widely used in agriculture as a biocontrol and plant growth promoting agent [106], has the potential to inhibit the mycelial growth of many fungal pathogens [48,107], including *P. expansum* [108], *M. fructicola* [109], and *B. cinerea* [110]. It produces a number of antifungal protein compounds, including lipopeptides, such as surfactin, iturin and fengycin [59], and antifungal hydolytic enzymes such as chitinase [111] and chitosanase [112]. However, in vitro assays did not detect any chitinase activity of the *B. subtilis* S57 strain selected in this study. All selected yeast and bacterial isolates produced at least two lytic enzymes, such as mannanase, cellulase, pectinase protease and chitinase, that were previously reported to be responsible for the antifungal activity of candidate BCAs [29,113,114,115,116,117]. However, there is no evidence based on the results of in vitro assays performed in this study that any of the enzymes tested may be regarded as a key factor of the strong antifungal activity shown in vitro by the three selected bacterial isolates. Another mechanism often involved in mycelium growth inhibition by BCAs is the production of antifungal VOCs [19,107,118]. In this study, VOCs produced by bacterial strains grown on PDA showed a higher inhibiting activity than those produced by yeast isolates. This is consistent with previous reports of other authors indicating that *Bacillus* species produce VOCs with a strong antifungal activity. Leeasuphakul et al. (2008) [48] demonstrated that VOCs produced by *B. subtilis* strains inhibited the mycelial growth of *P. digitatum* from 30 to 70%. Similarly, Arrebola et al. (2010) [103] reported COVs produced in vitro by *B. amyloliquefaciens* PPCB004 inhibited the mycelial growth of *P. crustosum*, *P. digitatum* and *P. italicum* by over 50, 3 and 25%, respectively. VOCs from another strain of *B. amyloliquefaciens* designated JBC36 inhibited the mycelial growth of *P. digitatum* and *P. italicum* by 57.8 and 54.1%, respectively [100]. Only in a few cases was the chemical nature of VOCS produced by *Bacillus* species identified [118]. The literature supports the hypothesis that, in this study, the stronger in vitro inhibitory effect of bacterial isolates on *P. digitatum* and *P. italicum* compared with yeast isolates could have been due prevalently to the difference in the chemical nature and amount of VOCs produced by *Bacillus* species. An additional mode of action that may contribute to the effectiveness of yeasts and bacteria as BCAs is the ability to form biofilms [32,119]. In a previous study, it was demonstrated that the formation of biofilms by *B. subtilis* is a complex process that includes the secretion of surfactin, a lipopeptide antimicrobial agent [119]. Several other studies have addressed the physiology and genetics of biofilm production by *Bacillus* species [120,121]. The knowledge of the role of biofilms in determining the performance of *B. amyloliquefaciens* and *B. subtilis* as BCAs is crucial, as diverse isolates of these two species have been selected and are already commercialized for this purpose. Interestingly, in this study, both selected yeast isolates, *C. oleophila* L12 and *D. hansenii* L16, showed an even higher capacity to form biofilms than that of *Bacillus* species. This could explain the efficiency of these two yeasts in suppressing Penicillium rot on artificially inoculated citrus fruits despite the relatively low antagonistic activity shown in vitro against *P. digitatum* and *P. italicum*. A similar correlation between the capability to form biofilms and efficiency in preventing Penicillium rot of citrus fruits was reported by Liu et al. (2019) [32] for other epiphytic yeasts. Conversely, differently from the results of this study, other researchers reported a strong in vitro inhibitory activity of selected yeast strains on the mycelium growth of *P. italicum* [122]. These discrepancies can be explained assuming that different mechanisms, including the production of killer toxins and non-protein inhibitory compounds, may be involved in the antagonistic efficiency of diverse candidate BCAs [123,124]. The ability to produce killer toxins has been reported for several candidate BCAs yeasts, including species of *Candida* and *Debaryomyces*, but may vary within the same species [125,126,127,128,129,130]. Consistently with the literature, this study confirmed that each of the five candidate BCAs exhibited more than a single inhibitory mechanism against test pathogens. Consequently, and not surprisingly, the efficacy in preventing infection on fruit did not strictly correlate with the in vitro inhibitory activity as the performance of a BCA depends on complex interactions pathogen/host plant/plant-associated microbiome/antagonist [92,131]. In particular, the isolate L12 of *C. oleophila* performed as a biocontrol agent better than expected on the basis of in vitro tests. This yeast and the *B. amyloliquefaciens* isolate were very effective in suppressing Penicillium rot in both wound-inoculated and naturally infected fruits. Their efficacy as candidate BCAs was comparable or even higher than that reported by other authors in previous studies [87,122,132]. Moreover, their application did not show any phytotoxic effect and did not impair the quality and shelf life of fruits. An aspect that would deserve to be investigated is the ability of bacterial and yeast isolates selected in this study to elicit the fruit defense response which is a common feature of several BCAs and natural substances [92,133,134]. The multifactorial nature of the efficacy of BCAs in preventing plant diseases is one of the advantages of biocontrol as it excludes the risk of pathogen resistance. Conversely, it might be a disadvantage, as many of these mechanisms are influenced by environmental conditions. This case was exemplified well by the bacterial isolate S15 of *B. amyloliquefaciens*, which performed better at a low temperature (4 °C), while its efficacy was significantly impaired at room temperature. This inconvenience may be circumvented by the use of broad-spectrum consortia of BCAs with complementary modes of action [135].

## 5. Conclusions

In conclusion, the yeast and bacterial isolates characterized in this study expand the list of promising BCAs that have the potential to be exploited commercially to control blue and green mold of citrus fruits. They could be formulated as either a single BCA or a consortium of diverse BCAs and applied alone or in mixture with natural substances as an alternative to conventional synthetic fungicides in ecofriendly management strategies of postharvest fruit decay.

## Figures and Tables

**Figure 1 jof-08-00818-f001:**
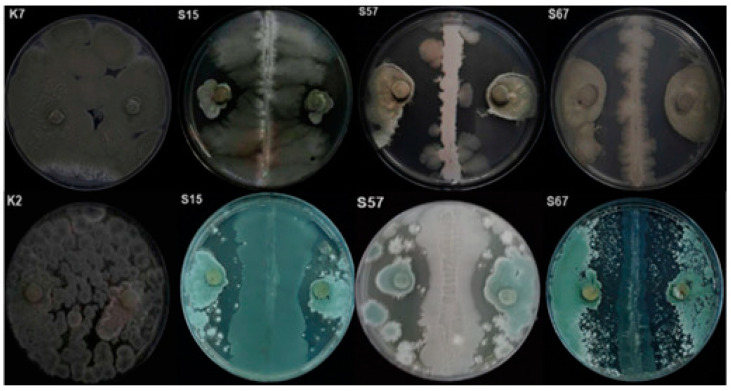
Dual culture test to evaluate the in vitro antagonistic activity of epiphytic bacteria (isolates S15, S57 and S67) against *Penicillium digitatum* and *P. italicum*. (K7) mycelial growth of *P. digitatum* without antagonist (control); (K2) mycelial growth of *P. italicum* without antagonist (control). Image taken after 7 days incubation at 25 °C.

**Figure 2 jof-08-00818-f002:**
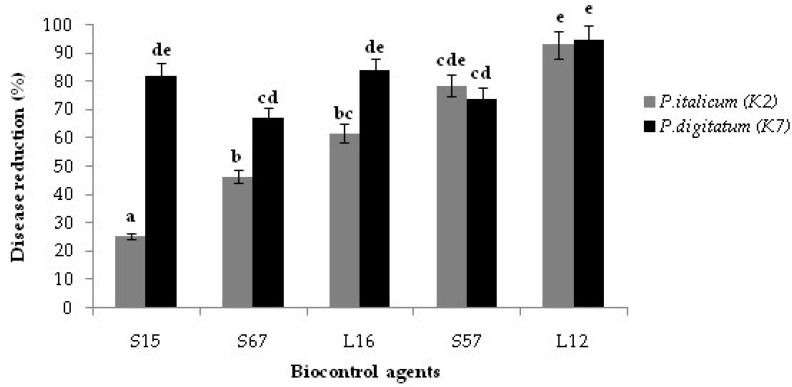
In vivo effectiveness of selected epiphytic yeast and bacterial isolates against *Penicillium digitatum* (K7) or *P. italicum* (K2): percentage of disease reduction on ‘Valencia late’ orange fruits after co-inoculation with the cell suspension (10^8^ CFU/mL) of yeasts (L12 and L16) or *Bacillus* species (S15, S57 and S67) and the conidial suspension (10^5^ conidia/mL) of *Penicillium* species. Columns with the same letters are not significantly different according to the Duncan’s test (*p* < 0.05).

**Figure 3 jof-08-00818-f003:**
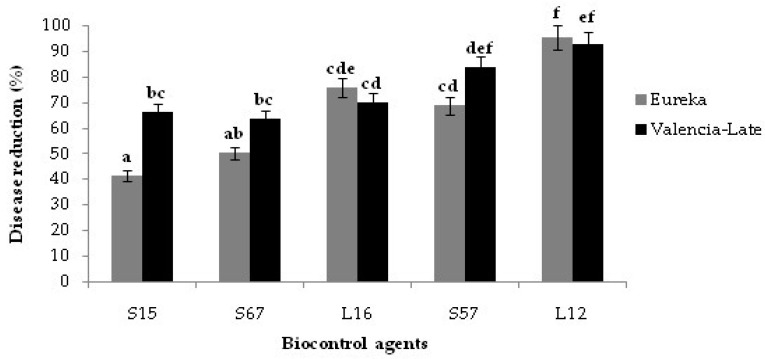
In vivo efficacy of selected epiphytic yeast (L12 and L16) and bacterial isolates (S15, S57 and S67): disease reduction on fruits of ‘Valencia late’ sweet orange and ‘Eureka’ lemon artificially inoculated with *Penicillium digitatum* and kept at ambient temperature for 7 days. Columns with the same letters are not significantly different according to the Duncan’s test (*p* < 0.05).

**Figure 4 jof-08-00818-f004:**
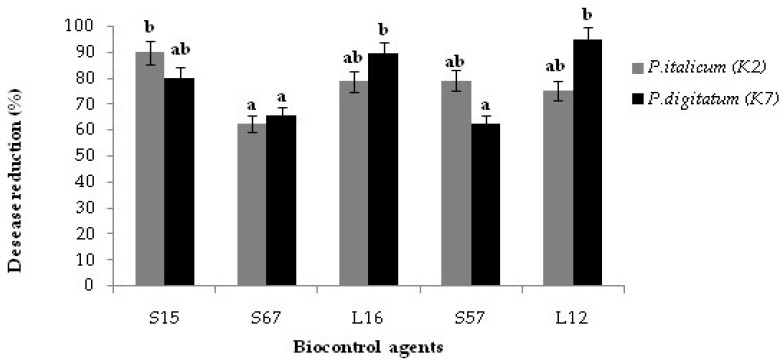
Effect of candidate BCAs yeast and *Bacillus* isolates on the percentage of disease reduction on wounded ‘Valencia Late’ orange fruits after co-inoculation with the suspension of yeasts or bacteria (10^8^ CFU/mL) and the conidial suspension (10^5^ conidia/mL) of *Penicillium digitatum* K7 or *P. italicum* K2. After inoculation, fruits were kept at +4 °C for 30 days. Columns with the same letters are not significantly different according to the Duncan’s test (*p* < 0.05).

**Figure 5 jof-08-00818-f005:**
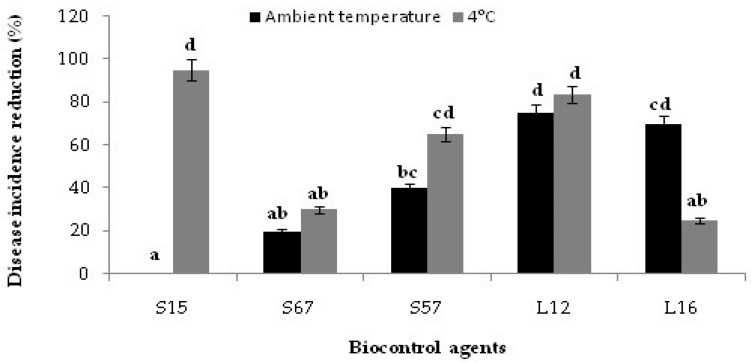
Reduction in green mold incidence on ‘Eureka’ lemon fruits treated with candidate BCAs yeasts and bacteria, artificially inoculated with *Penicillium digitatum* and stored at +4 °C for 30 days or at room temperature for 7 days. Columns with the same letters are not significantly different according to the Duncan’s test (*p* < 0.05).

**Figure 6 jof-08-00818-f006:**
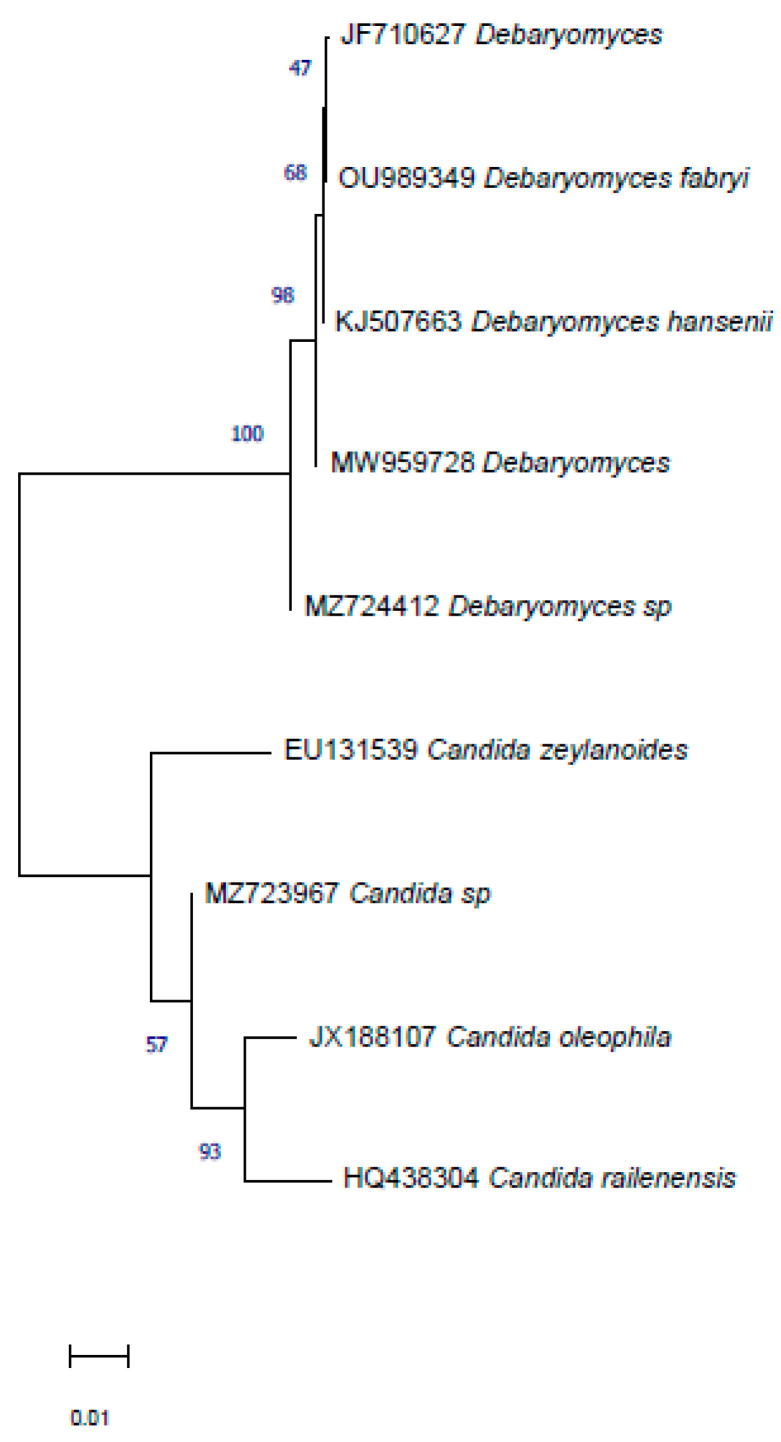
Phylogenetic tree based on internal transcribed spacer sequences (ITS) of rDNA, constructed using the neighbour-joining method with the software MEGA. Bars = 0.01 nucleotide substitutions per site.

**Figure 7 jof-08-00818-f007:**
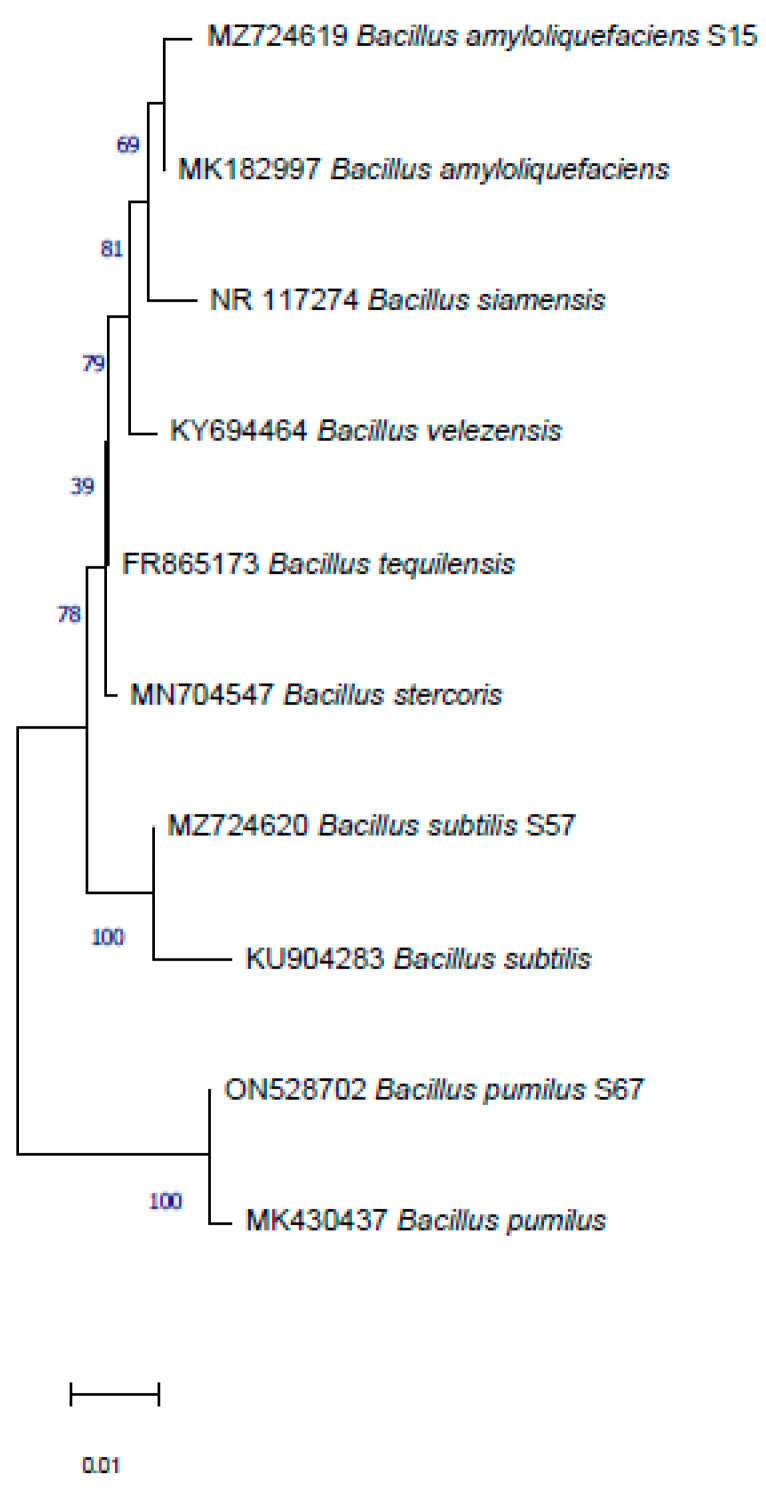
Phylogenetic tree based on 16S of rDNA of strains S15, S57 and S67, constructed using the neighbour-joining method with the software MEGA. Bars = 0.01 nucleotide substitutions per site.

**Figure 8 jof-08-00818-f008:**
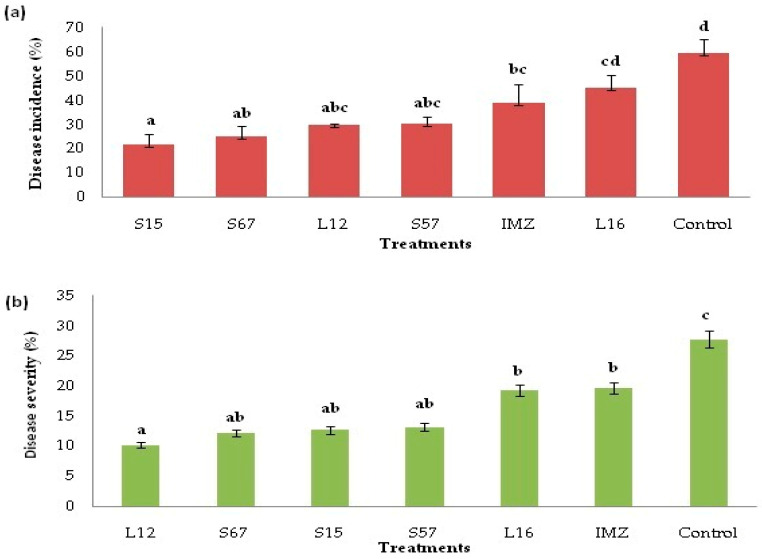
Effect of candidate BCAs epiphytic yeasts and bacteria (at concentration of 10^8^ cells/mL) on natural decay incidence (**a**) and disease severity (**b**) in ‘Valencia Late’ sweet orange fruits after storage at +4 °C and 95% RH for 90 days. S15, S57 and S67 indicate oranges treated with *Bacillus amyloliquefaciens*, *B. subtilis* and *B. pumilus*, respectively; L12 and L16 indicate oranges treated with, the yeasts *Candida oleophila* and *Debaryomyces hansenii*, respectively; IMZ indicates oranges treated with imazalil (2 mL/L a.i.); Control includes oranges without any treatment. Columns with different letters are significantly different according with the Duncan’s multiple range test, (*p* < 0.05).

**Figure 9 jof-08-00818-f009:**
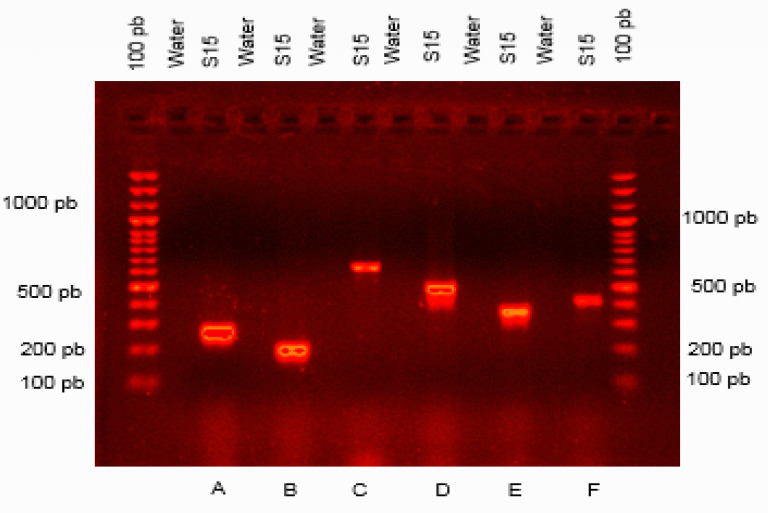
PCR amplification products of antibiotic biosynthetic genes of *Bacillus amyloliquefaciens* (S15), using primers for amplification of genes encoding (A) fengycin (FENDF/R) 269 bp; (B) surfactin (SRFAF/R) 201 bp; (C) iturin A (ITUD1F/R) 647 bp; (D) bacilysin (BCAF/R) 498 bp; (E) bacillomycin (BMYBF/R) 370 bp; and (F) mycosubtilin (Am1F/Tm1R) 419 pb.

**Table 1 jof-08-00818-t001:** In vitro antagonistic activity of selected bacteria and yeast isolates on mycelial growth of *Penicillium digitatum* and *P. italicum*, determined by the dual culture test and expressed as % inhibition of mycelium growth.

Isolates	*P. digitatum* K7 Mycelial Growth Inhibition (%)	*P. italicum* K2 Mycelial Growth Inhibition (%)
*Bacillus amyloliquefaciens* S15	76 ± 2 a	45 ± 2 b
*B. subtilis* S57	60 ± 0 b	73 ± 2 a
*B. pumilus* S67	60 ± 0 b	32 ± 7 c
*Candida oleophila* L12	32 ± 5 c	20 ± 6 c
*Debaryomyces hansenii* L16	28 ± 5 c	20 ± 6 c

Values (±SD) followed by the same letters are not significantly different according to Duncan’s multiple range test (*p* < 0.05).

**Table 2 jof-08-00818-t002:** Effect of biocontrol strain on postharvest quality of citrus fruit stored at +4 °C for 90 days.

Treatment ^ab^	Weight Loss (%)	Fruit Firmness Variation (%)	TSSs Variation (%)	TA Variation (%)	Ascorbic Acid Variation (%)
S57	22.63 a	49.27 bc	5.05 g	31.82 b	2.90 a
IMZ	24.14 b	29.200 ab	1.56 f	40.37 d	7.40 d
S15	25.98 c	29.03 ab	−4.66 a	36.62 c	3.65 b
L12	26.80 d	11.57 a	0.30 c	77.03 g	2.72 a
S67	27.36 d	33.61 ab	0.45 e	45.53 e	3.84 b
L16	29.19 e	61.71 c	0 b	0.72 a	2.90 a
Control	30.43 f	31.19 ab	0.33 d	61.73 f	4.76 c

**^a^** Weight loss%; N: Fruit firmness (%); TSSs: Total soluble solids (%); TA: Titrable acidity (%); Vit C: Ascorbic acid. Biocontrol strains cell suspension of *Bacillus amyloliquefaciens* (S15), *B. subtilis* (S57), *B. pumilus* (S67), *Debaryomyces hansenii* (L16) and *Candida oleophila* (L12). IMZ: indicated imazalil at 2 mL/L; Control: indicated non-treated fruits. **^b^** Values with the same letters in a column are not significantly different according to Duncan’s multiple range test (*p* < 0.05).

**Table 3 jof-08-00818-t003:** Lytic enzyme activity of selected yeasts and *Bacillus* isolates as determined by spot-inoculation in Petri dishes.

Strains	Enzymatic Activity						
	Mananase	Amylase	Protease	Cellulase	Chitinase	Pectinase	Urease
S15	+	+	+	+	+	+	-
S57	+	+	+	+	-	+	-
S67	+	-	+	-	-	-	-
L12	+	-	-	+	-	-	-
L16	+	-	-	+	-	+	-

Note: (+) positive reaction as shown by the presence of a clear halo around the colonies (-) and negative reactions indicating that the isolates had no lytic enzyme activity.

**Table 4 jof-08-00818-t004:** In vitro antagonistic activity of volatile organic compounds (VOCs) produced by candidate BCAs yeasts and bacteria toward *Penicillium digitatum* (K7) and *P. italicum* (K2) as expressed in terms of mycelial growth inhibition (% colony diameter reduction compared to the control).

Isolates	% Reduction of *P. digitatum* K7 Colony Diameter	% Reduction of *P. italicum* K2 Colony Diameter
*Bacillus amyloliquefaciens* S15	61 ± 0.01 a	54 ± 0.00 a
*B. subtilis* S57	60 ± 0.00 a	51 ± 0.05 a
*B. pumilus* S67	47 ± 0.02 b	51 ± 0.01 a
*Debaryomyces hansenii* L16	41 ± 0.01 c	6 ± 0.00 c
*Candida oleophila* L12	7 ± 0.05 d	41 ± 0.01 b

Note: Means (±SE) followed by different letters are significantly different according to Duncan’s multiple range test (*p* < 0.05).

**Table 5 jof-08-00818-t005:** Ability of five selected candidate BCAs to form biofilms.

Isolates	Optical Density (OD)
*Bacillus amyloliquefaciens* S15	0.20 ± 0.35 c
*B. subtilis* S57	0.26 ± 0.39 c
*B. pumilus* S67	0.29 ± 0.00 c
*Candida oleophila* L12	0.91 ± 0.28 a
*Debaryomyces hansenii* L16	0.75 ± 0.24 b

Note: OD values are directly correlated with the capacity of yeast and bacterial cells to adhere to the polystyrene dishes. Different letters indicate significantly different values according to Duncan’s multiple range test (*p* < 0.05).

## Data Availability

Not applicable.

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
