# Peer review of "Epiphytic Yeasts and Bacteria as Candidate Biocontrol Agents of Green and Blue Molds of Citrus Fruits"

_jof, 2022, doi:10.3390/jof8080818_

Round 1
Reviewer 1 Report
The details of revision has been performed in the attached pdf file.

Author Response
Reviewer 1
Review Report (Round 1)
Comment 1: p. 1, line 25: It is not important to mention these details in the abstract.
Correction: at either low (4°C) or ambient temperature Deleted from the abstract
Comment 2 : p. 1, line 29: why you call "antifungal lipopeptides" ?
Correction: antifungal lipopeptides Its known that lipopeptides especially the members of the iturin family, including iturin A, bacillomycin, mycosubtilin and surfactin families have been reported to possess strong antifungal activity against a wide range of fungi, including plant pathogens. They are able to modify the permeability and lipidic composition of fungal membrane.
Comment 3: p. 1, line 40: [2,3,4]
Correction: [2,3,4] [2-4]
Comment 4: p. 2, line 62, 63: These drawbacks of synthetic fungicides have prompted researchers to seek alternative and more safe methods for controlling postharvest decay of fruits [6,18-21].
Correction 4: Two other papers are cited.
[22] Camele, I.; Elshafie, H.S.; Caputo, L.; Sakr, S.H.; De Feo, V. Bacillus mojavensis: Biofilm formation and biochemical investigation of its bioactive metabolites. J. Biol. Res. 2019, 92.
[23] Elshafie, H.S.; Caputo, L.; De Martino, L.; Gruľová, D.; Zheljazkov, V.Z.; De Feo, V.; Camele, I. Biological investigations of essential oils extracted from three Juniperus species and evaluation of their antimicrobial, antioxidant and cytotoxic activities. J. Appl. Microbiol. 2020, 129, 1261-1271.
These drawbacks of synthetic fungicides have prompted researchers to seek alternative and more safe methods for controlling postharvest decay of fruits [6,18-23].
Comment 5: p. 3, line 124: write briefly some details of this assay, how did you carry out the assay ?.
Correction 5: In vitro antagonism assays against P. digitatum and P. italicum were performed using the dual-culture test described by [63] Each yeast or bacterium was applied as a straight line passing through the center of 9 cm Petri dish containing PDA medium. Then, a mycelium plug of 5 mm in diameter of the pathogen (P. digitatum or P. italicum) was placed approximately 2.5 cm away from each side of the tested strain.
Comment 6: p. 4, line 155,156,157,158,159,160,161,162,163,164,165,166: it is too much explanation under the formula. Write these details in the text
Correction 6: Details mentioned in the text.
Comment 7: p. 4, line 167: write the deatils of the statistical analysis
Correction 7 : Statistical analysis was performed using the program Stat soft. Inc. 2011. STATISTICA (data analysis software system) version 10. www.statsoft.com. An analysis of variance and mean comparison test were performed using Duncan’s test with 5 % significance level.
Comment 8: p. 17, line 589,590,591,592 : in the conclusion, it is not advisable to add references
Correction 8 : [51,73,92] Deleted from the conclusion.
In conclusion, the yeast and bacterial isolates characterized in this study expand the list of promising BCAs that have the potential to be exploited commercially to control blue and green mold of citrus fruits. They could be formulated as either a single BCA or a consortium of diverse BCAs and applied alone or in mixture with natural substances as an alternative to conventional synthetic fungicides in ecofriendly management strategies of postharvest fruit decay.

Reviewer 2 Report
It is of great interest to screen new biocontrol agents for the postharvest decay of citrus fruits, the present study isolated more than 180 yeasts and bacteria from citrus peel, and found that two yeast and three bacteria have inhibitory activities against Penicillium digitatum and P. italicum. The evidence consists of in vitro and in vivo data. Further, the authors performed experiments to probe the potential mechanism, and found that multiple modes of action were involved, including the ability to form biofilms and produce antifungal lipopeptides, lytic enzymes, and volatile compounds. The present study was interesting and meaningful, the data could support the conclusion although the overall research is somehow preliminary.
The writing and expression in the present manuscript, and needs substantial improvement.
Author Response
Reviewer 2
Review Report (Round 1)
The manuscript has been slightly improved.
